# Facile Synthesis of High Performance Iron Oxide/Carbon Nanocatalysts Derived from the Calcination of Ferrocenium for the Decomposition of Methylene Blue

**Thinnaphat Poonsawat, Thanyaphat Techalertmanee, Peerapong Chumkaeo, Isti Yunita, Titiya Meechai, Montree Namkajorn, Soraya Pornsuwan and Ekasith Somsook ***

NANOCAST Laboratory, Center for Catalysis Science and Technology (CAST), Department of Chemistry and Center of Excellence for Innovation in Chemistry, Faculty of Science, Mahidol University, 272 Rama VI Rd., Ratchathewi, Bangkok 10400, Thailand; p.thinnaphat@gmail.com (T.P.); tthanyaphat@gmail.com (T.T.); nanocast.pck@gmail.com (P.C.); isti_yunita@yahoo.com (I.Y.); titiyasouy@gmail.com (T.M.); montree_072@hotmail.com (M.N.); soraya.por@mahidol.edu (S.P.)

* Correspondence: ekasith.som@mahidol.ac.th; Tel.: +66-22015123; Fax: +66-23547151

**Abstract:** Iron oxide/carbon nanocatalysts were successfully synthesized by the calcination of ferrocenium at high temperatures ranging from 500 to 900 °C. Then the synthesized nanocomposites were characterized by XRD (X-Ray Diffraction), TEM (Transmission Electron Microscopy), VSM (Vibrating-Sample Magnetometry), BET (Brunauer-Emmett-Teller surface area measurements), TGA (Thermogravimetric Analysis), XPS (X-Ray Photoelectron Spectroscopy), EPR (Electron Paramagnetic Resonance), and CHN elemental analysis. The prepared nanocatalysts were applied for the decomposition of methylene blue as a model in wastewater treatment. It was unexpected to discover that the prepared nanocatalysts were highly active for the reaction with methylene blue in the dark even though no excess of hydrogen peroxide was added. The nanocatalyst calcined at 800 °C exhibited the rod shape with the best catalytic activity. The nanocatalysts could be reused for 12 times without the significant loss of the catalytic activity.

**Keywords:** iron oxide/carbon; ferrocenium; decomposition; methylene blue

## 1. Introduction

The increasing population and rapid growth of urbanization has led to the increasing shortage of clean water and a higher demand for wastewater treatment [1,2]. Many methods such as chemical oxidation, coagulation, flotation, reverse osmosis, photochemical degradation, membrane filtration, ozonation, electrochemical treatment, and adsorption have been used to eliminate organic and inorganic components from wastewaters [3–6]. Methylene blue (MB) is an important basic dye widely used in textile and paper industries [7,8]. This dye leads off eye burns, breathing disorders, heart rate increases, shock, cyanosis, jaundice, quadriplegia, tissue necrosis, nausea, vomiting, mental confusion, painful micturition, and methemoglobinemia [9]. In this work, MB was selected as a representative to be removed from wastewater.

Advanced oxidation processes have been developed as innovative tools involving in the highly reactive oxygen species for the treatment of wastewater [10–14]. The Fenton reaction [15] is a reaction between Fe (II) and hydrogen peroxide that generates Fe (III) hydroxide and hydroxyl radicals, and is commonly studied for the Fenton-like reagents (Fe (III)) [16–18] and photocatalysts (TiO$_2$) [19–24], and it can also generate hydroxyl radicals to eliminate pollutants. Iron(III) or iron oxide catalysts were

widely used for Fenton-like degradation or decolorization of methylene blue, such as $SnO_2/Fe_2O_3$ [25], $\gamma$-$Fe_2O_3$ nanocrystals-anchored macro/meso-porous graphene [26], zinc–iron mixed oxide/carbon nanocomposites [27], ferrocenated compounds [28,29], Fe-doped $Sr_2Bi_2O_5$ [30], $Fe_3O_4$–wheat straw [31], iron oxide ($Fe_3O_4$, $\gamma$-$Fe_2O_3$, $\alpha$-$Fe_2O_3$)/cellulose [32], $\alpha$-$Fe_2O_3/TiO_2$ [33], $\alpha$-$Fe_2O_3$/MCM-41 [34], Fe (II)Fe (III)-LDHs [35], $ZrFe_2O_5$ [36], and $\alpha$-$Fe_2O_3/Bi_2MoO_6$ [37]. Recently, the paper mill sludge-derived magnetically separable heterogeneous catalyst for the Fenton-like reaction by degradation of MB was studied; the Fe-loaded sludge was calcined in air at 380 °C for 2 h into the paper mill sludge-derived Fe-loaded nanocomposite. These catalysts can show reusability and stability in five repeated runs [38]. However, these catalysts for the degradation of MB required the use of $H_2O_2$ as an oxidant. Hydroxyl radicals are the key intermediates in those processes. Even though photocatalysts are attractive, they require light irradiation and also high solar energy conversion. Therefore, new catalysts with improved activities will be needed for sustainable development.

Ferrocene [39] is a redox active species that can react with hydrogen peroxide to generate hydroxyl radicals for the oxidation of organic compounds [40,41]. In addition, ferrocene has been used as a precursor for the synthesis of iron oxide nanoparticles and carbon nanotubes [42–44]. Iron oxides are stable, abundant and available on earth and very attractive for catalysis [45–47]. Herein, novel ferrocene-derived iron oxides/carbon nanocatalysts showing high activities for the MB decomposition in the dark condition without the addition of $H_2O_2$ will be demonstrated. In addition, the catalysts were characterized by transmission electron microscopy (TEM), X-ray photoelectron spectroscopy (XPS), X-ray powder diffraction (XRD), electron paramagnetic resonance (EPR) spectroscopy, Brunauer-Emmett-Teller (BET) surface area analysis, thermogravimetric analysis (TGA), vibrating sample magnetometer (VSM) and carbon, hydrogen and nitrogen (CHN) elemental analysis. The products or MB solutions were analyzed by Ultraviolet-Visible (UV-VIS) spectroscopy, flame atomic absorption spectroscopy (FAAS), and electrospray ionization mass spectroscopy (ESI–MS).

## 2. Results and Discussion

Ferrocenium $[(C_5H_5)_2Fe]^+$ was simply prepared by the reaction of ferrocene with concentrated sulfuric acid [39,48,49]. Then the calcination of ferrocenium in a closed crucible was carried out in a furnace under atmospheric condition at a specific temperature ranging from 500–900 °C as shown in Scheme 1.

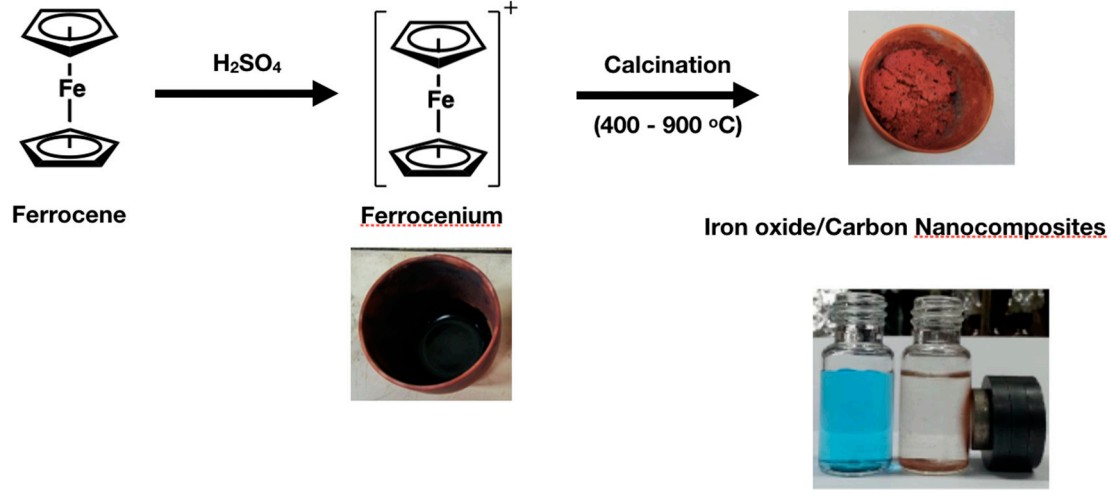

**Scheme 1.** The preparation of iron oxides/carbon nanocomposites.

The XRD patterns at higher temperatures (500–900 °C) showed the distinctive pattern of $\alpha$-$Fe_2O_3$ as shown in Figure 1. As the calcination temperature increased, the XRD intensities became more

distinctive showing the hexagonal phase of $\alpha$-Fe$_2$O$_3$ (JCPDS no. 33-0664) [50,51]. $\alpha$-Fe$_2$O$_3$ is known as an active catalyst for the photocatalytic activity [52–54]. Moreover, the average crystallite sizes in the samples were calculated from the XRD line broadening using the Scherrer's formula [55]. The results of 400, 500, 600, 700, 800 and 900 °C were 26, 47, 56, 59, 67 and 67 nm, respectively. As shown in Figure 2 (TEM), it was obviously that the prepared samples were nanocomposites in which the particle sizes of samples were 100, 66, 28, 28 and 43 nm obtained from the calcination at 400, 500, 600, 700, and 900 °C, respectively. In addition, the 800 °C sample exhibited a rod shape with 555 nm length and 60 nm width. The particle size of the prepared nanocomposites was smallest for the calcination at 600 and 700 °C. Therefore, calcination temperatures affect the size of the particles, which increases the temperature 400–700 °C to a small particle size and high temperatures of 800–900 °C to a large particle size due to particle agglomeration.

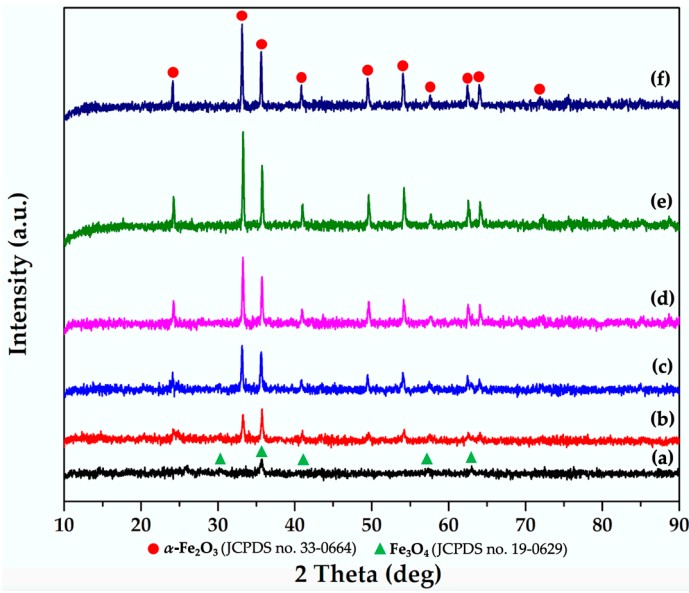

**Figure 1.** XRD patterns of nanocomposites synthesized by the calcination of ferrocenium at different temperatures. (**a**) 400 °C, (**b**) 500 °C, (**c**) 600 °C, (**d**) 700 °C, (**e**) 800 °C, and (**f**) 900 °C.

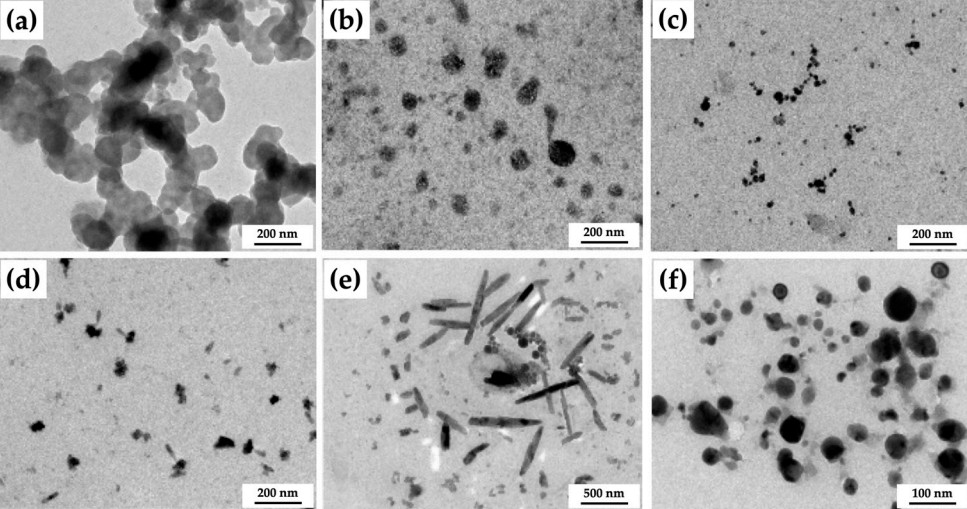

**Figure 2.** TEM images of nanocomposites synthesized at (**a**) 400 °C, (**b**) 500 °C, (**c**) 600 °C, (**d**) 700 °C, (**e**) 800 °C, and (**f**) 900 °C.

Previously, the pyrolysis of ferrocene was performed at high temperature (1050 °C) in a closed reactor in a sophisticated furnace under an inert atmosphere to yield a mixture of carbon and iron [51]. Then the oxidation of the mixture was carried out under a constant air flow to obtain carbon-coated $\alpha$-Fe$_2$O$_3$. In our experiment, a simple experiment to produce iron oxide from ferrocene was successfully carried out by the transformation of ferrocene to ferrocenium first, and then the calcination of the synthesized ferrocenium in a closed crucible inside a furnace under ambient atmosphere. The ionic character of ferrocenium in the mixture enhanced the thermal stability in which the mixture started to be decomposed at 500 °C and completed the decomposition of organic moieties at 600 °C as shown in Figure 3 and Table 1. The prepared nanocomposites exhibited the magnetic properties using VSM analysis as shown in Figure 4 and Table 2. It can be found that the magnetic saturation (Ms) values of 500–900 °C were 4.43, 6.71, 4.65, 5.86, and 0.28 emu g$^{-1}$, respectively. The 500–800 °C samples exhibited super-paramagenetic behavior that observed the narrower loop, while the broad loop at 900 °C showed the ferromagenetic behavior. The difference between the ferromagenetic behavior and super-paramagenetic behavior is primarily determined by the size of the particle. CHN analysis provided the carbon percentages at 25.1, 25.8, 28.2, 6.5, and 1.5 for the samples calcined at 500, 600, 700, 800, and 900 °C, respectively, and the hydrogen percentages were low for all samples.

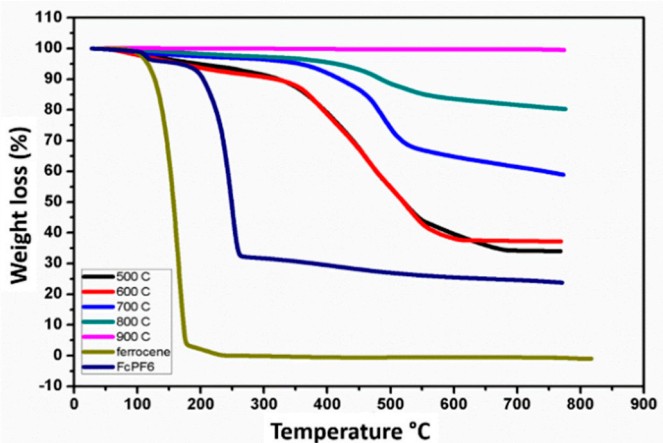

**Figure 3.** TGA curves of nanocatalysts synthesized at different calcination temperatures including ferrocene and ferrocenium salt.

**Table 1.** Show weight lost in percentage and in milligram in three parts.

| Samples | Part I (350–450 °C) | | Part II (450–550 °C) | | Part III (550–800 °C) | | Sample Weight (mg) |
|---|---|---|---|---|---|---|---|
| | Weight Loss (%) | Weight Loss (mg) | Weight Loss (%) | Weight Loss (mg) | Weight Loss (%) | Weight Loss (mg) | |
| 500 °C | 5.675 | 0.376 | 60.293 | 3.997 | 34.032 | 2.256 | 6.630 |
| 600 °C | 8.127 | 0.548 | 54.717 | 3.693 | 37.156 | 2.508 | 6.750 |
| 700 °C | 2.980 | 0.199 | 37.962 | 2.535 | 59.058 | 3.943 | 6.678 |
| 800 °C | 2.249 | 0.145 | 17.360 | 1.125 | 80.391 | 5.212 | 6.484 |
| 900 °C | 0.386 | 0.028 | 0.188 | 0.013 | 99.426 | 7.280 | 7.323 |

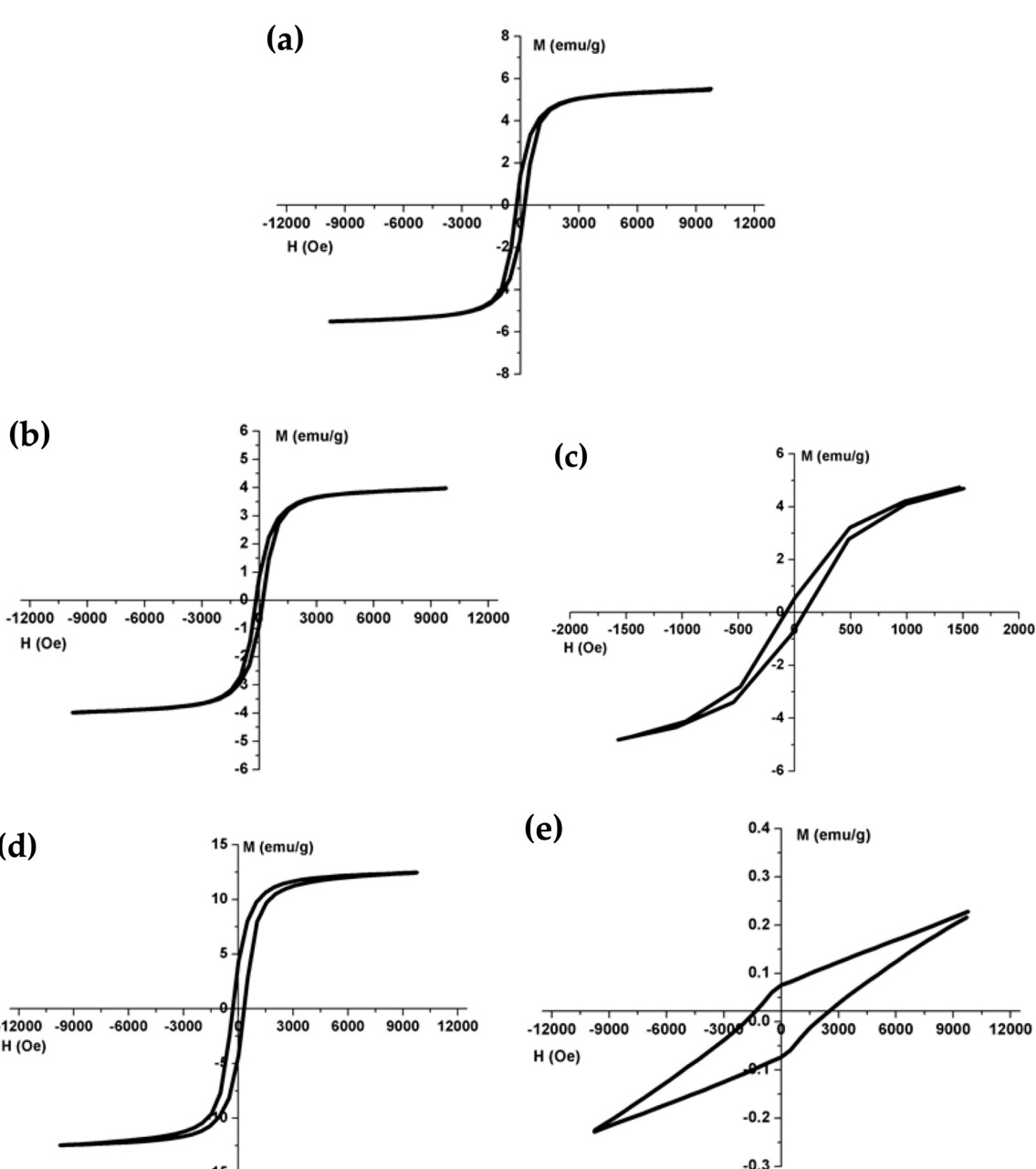

**Figure 4.** Hysteresis curve of nanocatalysts synthesized at different calcination temperatures. (**a**) 500 °C, (**b**) 600 °C, (**c**) 700 °C, (**d**) 800 °C, and (**e**) 900 °C.

**Table 2.** Shows the coercivity ($H_{ci}$), magnetization ($M_s$), and retentivity ($M_r$).

| Samples | Coercivity (G) | Magnetization (emu/g) | Retentivity (emu/g) |
|---|---|---|---|
| 500 °C | 198.73 | 4.4323 | 1.1041 |
| 600 °C | 45.677 | 6.7054 | 0.35029 |
| 700 °C | 48.933 | 4.6540 | 0.24313 |
| 800 °C | 238.11 | 5.8552 | 1.6709 |
| 900 °C | 3878.3 | 0.27934 | 0.11349 |

The XPS spectra of nanocomposites also confirmed the presence of iron oxide and carbon in the samples as shown in Figure 5. Considering the XPS spectra of Fe 2p from nanocomposites calcined at 500–900 °C (Table 3), there was a tendency to detect species at lower binding energies for the calcination at 600 °C, indicating the higher ratio of iron (II) and iron (III) [56–59]. The presence of the iron (II)

species was probably derived from the reduction of iron (III) by carbon, indicating the close contact of iron oxide and carbon in the nanocomposites. The C 1s spectrum at the binding energies were assigned to the C–C, C–O, O–C=O, and O–C=OH, shown in Figure 5. The $Fe_3C$-based materials were potential Fenton-like catalysts [60], that was found at 283.8 eV in the 700 and 800 °C nanocatalysts. The curves fitting of O 1s for 500–900 °C nanocatalysts found the binding energies at 530 eV that were assigned to Fe–O, while the 700 and 800 °C nanocatalysts were $OH^-$ of FeOOH (530.8 eV) and Fe–OH (532 eV).

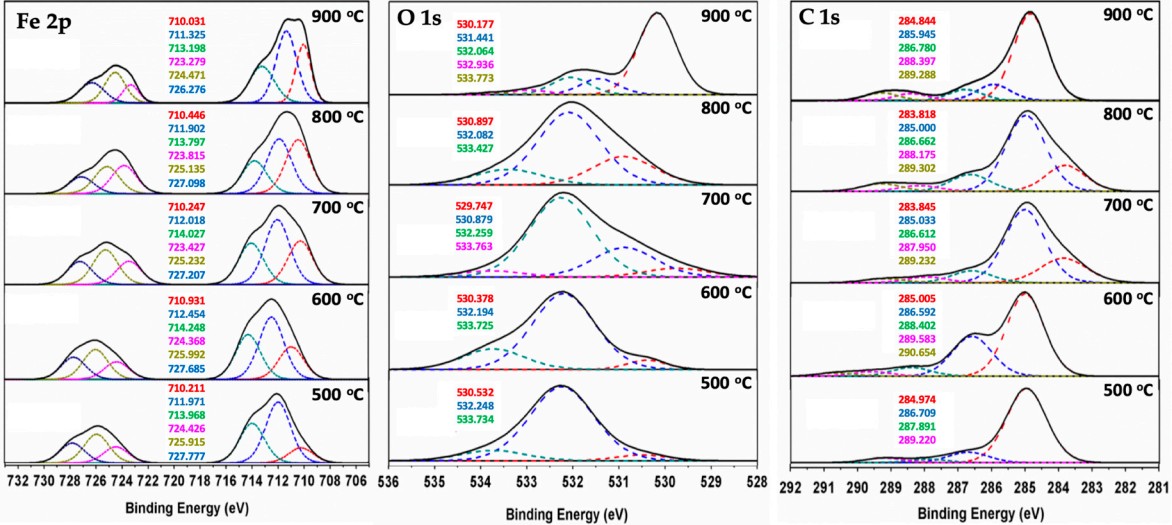

**Figure 5.** XPS spectrum of Fe 2p, O 1s, and C 1s of nanocomposites synthesized at different calcination temperatures.

**Table 3.** The XPS assignments from binding energies.

| Binding Energy (eV) | | | | | Assignments |
|---|---|---|---|---|---|
| **500 °C** | **600 °C** | **700 °C** | **800 °C** | **900 °C** | |
| - | - | - | - | 710.0 | $Fe^{2+}2p_{3/2}$ of FeO |
| 710.2 | 710.9 | 710.4 | 710.4 | 711.3 | $Fe^{3+}2p_{3/2}$ of $Fe_2O_3$ |
| 711.9 | 712.4 | 711.9 | 711.9 | - | $Fe^{3+}$ of FeOOH |
| 713.9 | 714.2 | 713.7 | 713.7 | 713.2 | $Fe^{2+}2p_{3/2}$ of $Fe_2O_3$ |
| - | - | - | - | 723.3 | - |
| 724.9 | 724.4 | 723.8 | 723.8 | 724.4 | $Fe^{3+}2p_{1/2}$ of $Fe_3O_4$ |
| 725.9 | 725.9 | 725.1 | 725.1 | 726.3 | $Fe^{3+}2p_{1/2}$ of $Fe_2O_3$ |
| 727.7 | 727.7 | 727.1 | 727.1 | - | $Fe^{3+}2p_{1/2}$ of $Fe_2O_3$ |
| - | - | 529.7 | - | 530.1 | Fe–O |
| 530.5 | 530.4 | - | - | - | Fe–O |
| - | - | 530.8 | 530.8 | - | –OH of FeOOH |
| 532.2 | 532.2 | - | - | 531.4 | O=C |
| - | - | 532.2 | 532.1 | 532.0 | Fe–OH |
| - | - | - | 533.4 | 532.9 | O–C |
| 533.7 | 533.7 | 533.7 | - | 533.7 | C–OH |
| - | - | 283.8 | 283.8 | - | $Fe_3C$ |
| 284.9 | 285.0 | 285.0 | 285.0 | 284.8 | C–C |
| - | - | - | - | 285.9 | C–O |
| 286.7 | 286.6 | 286.6 | 286.6 | 286.7 | O–C=O |
| 287.9 | - | 287.9 | - | - | C=O |
| - | 288.4 | - | 288.2 | 288.4 | O=C–OH |
| 289.2 | 289.6 | 289.2 | 289.3 | 289.3 | O=C–OH |
| - | 290.6 | - | - | - | $CO_3^{2-}$ |

The decomposition of MB was selected for testing the catalytic activity of iron oxide/carbon nanocomposites. First, 100 cm$^3$ of $1 \times 10^{-5}$ mol/dm$^3$ of methylene blue solution was added into a container and then 100 mg of nanocatalyst was added. The mixture became acidic at pH of 3. It was allowed to be stirred for 5 min for adsorption and desorption equilibrium. The solution was sampled every 10 min and the percentages of the decomposition of the solution were calculated after being measured by UV-VIS spectroscopy by monitoring the absorbance at 662 nm, as shown in Figure 6.

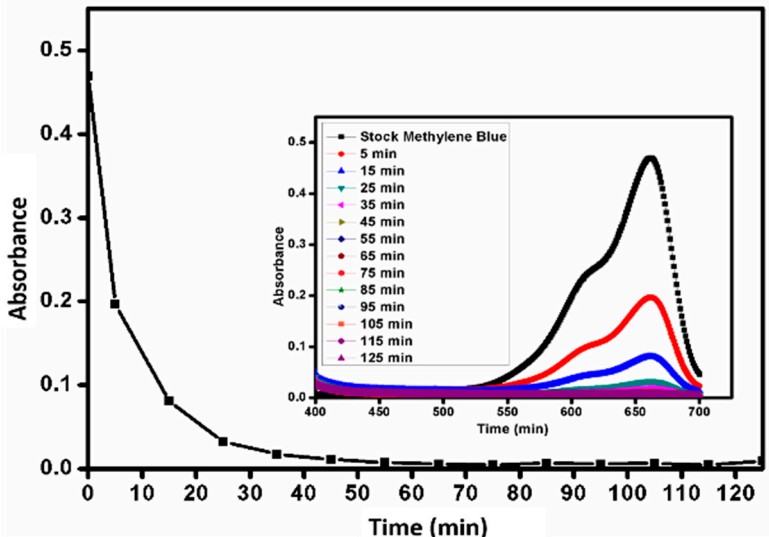

**Figure 6.** The absorbance of methylene blue at $\lambda_{max}$ 662 nm and the reaction time. The inset was visible spectra of methylene blue at different reaction times.

First, the decomposition of methylene blue in the presence of iron oxide/carbon nanocomposites was carried out under UV irradiation and hydrogen peroxide was added to the reaction. However, the color of methylene blue disappeared suddenly when hydrogen peroxide was added. As shown in Figure 7, the control experiment showed the inactive decomposition. The iron oxide/carbon nanocomposites synthesized by the calcination of ferrocenium at 500–800 °C were very highly active nanocatalysts for the decomposition of methylene blue, even though no UV irradiation was applied. However, the catalytic activity of iron oxide/carbon nanocomposites in the dark was slightly lower than the catalytic activity under UV irradiation. The most active nanocatalysts were nanocomposites calcined at 800 °C, which exhibited the rod shape of particle size, specific surface area 49.5 m$^2$/g (Table 4) and the high concentration of iron (II) in the sample while the nanocomposites calcined 900 °C exhibited the low activities. The 500–700 °C nanocatalysts showed the comparable percentage of carbon while the 800 and 900 °C nanocatalysts showed very low percentage of carbon. The decomposition of methylene blue was very slow and incomplete in the presences of nanocatalysts without carbon. However, the low percentage of carbon in the 800 °C nanocatalyst showed a significant increasing of the catalytic activity. Therefore, this indicated the synergistic activity of iron oxide and carbon in the decomposition of methylene blue. The other catalysts such as Fe$^0$/Fe$_3$O$_4$ [61], iron oxide/silica [62,63], iron oxide/MCM-41 [34], and iron-based or iron oxide/carbon nanocomposites [64–66], have been reported for the decomposition of dyes with the excessive amount of hydrogen peroxide while the other catalysts such as Cu$_2$(OH)$_2$NO$_3$/ZnO [67], pyrite (FeS$_2$) [68], and Fe/Fe$_2$O$_3$ [69,70] have also been reported for the effective decomposition of dyes without the excess of hydrogen peroxide.

As shown in Figure 8, different amounts of nanocatalysts exhibited different catalytic activities. The decomposition efficiency is increased with the increasing amount of catalyst. This indicated the active species must be derived from nanocatalysts, not from other sources. However, the catalytic activity could not be differentiated at the higher amount of nanocatalysts (more than 100 mg) due to probably the limiting diffusion of oxygen gas. By assuming the pseudo-first-order reaction of the

decomposition of methylene blue, the decomposition rate constants may be able to be extracted from these curves. A linear relationship between the decomposition rate constants and the amount of nanocatalysts was obtained indicating that the active species must be derived from nanocatalysts (see Figure 9).

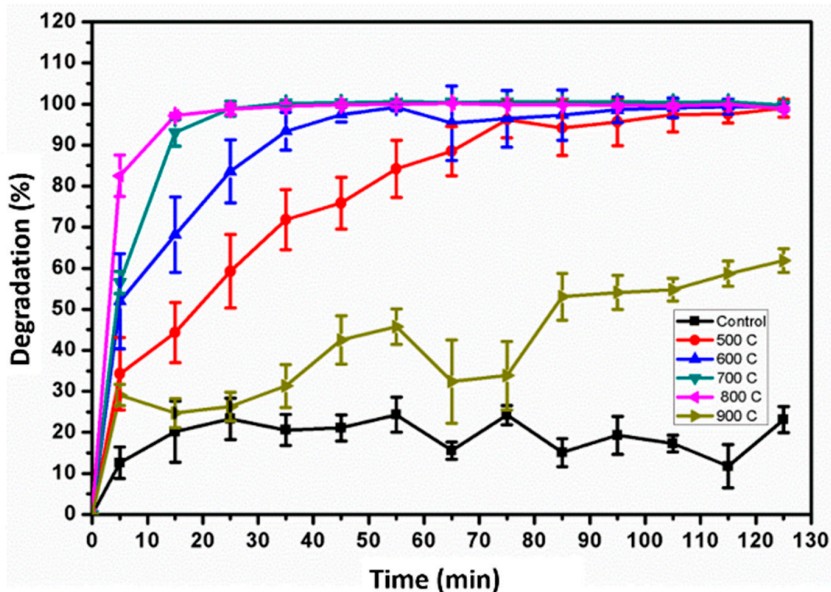

**Figure 7.** The decomposition of methylene blue under the dark condition over nanocatalysts synthesized at different calcination temperatures.

**Table 4.** Showing the analysis of surface area of nanocatalysts synthesized at different calcination temperatures.

| Samples | Specific Surface Area (m$^2$/g) | Total Pore Volume (cc/g) | Pore Size (nm) |
|---|---|---|---|
| 500 °C | 210.5 | $2.107 \times 10^{-1}$ | 400.3 |
| 600 °C | 299.5 | $2.806 \times 10^{-1}$ | 374.9 |
| 700 °C | 280.4 | $2.804 \times 10^{-1}$ | 400.1 |
| 800 °C | 49.51 | $1.233 \times 10^{-1}$ | 996.4 |
| 900 °C | 7.898 | $1.144 \times 10^{-2}$ | 579.5 |

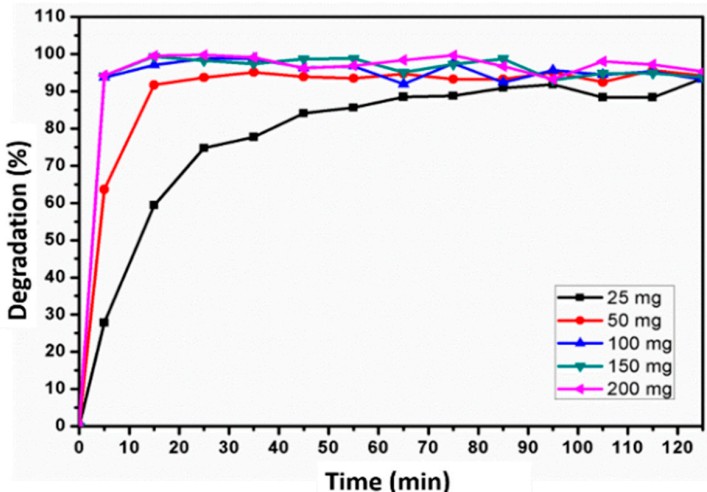

**Figure 8.** The decomposition of methylene blue under the dark condition over different amounts of the 800 °C nanocatalysts.

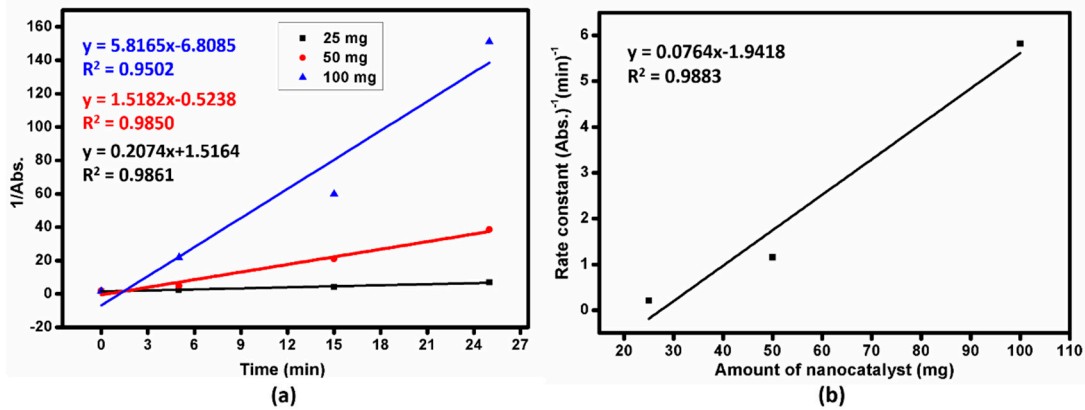

**Figure 9.** (**a**) The second order kinetic studies of methylene blue decomposition in the presence of iron oxide/carbon nanocomposites at different amounts; (**b**) plot of rate constant and the amount of nanocatalysts.

As shown in Figure 10, the EPR spectra of DMPO adducts in the presence of nanocatalysts confirmed that the active species was superoxide radicals (see Table 5) for EPR parameters simulated by WinSim [71]. However, DMPO-OOH has very short lifetime and it can readily decompose to hydroxyl radical. Therefore, the EPR signals of DMPO-OH adduct were also observed. Moreover DMSO is an inhibitor which can react with DMPO to give EPR signals [72].

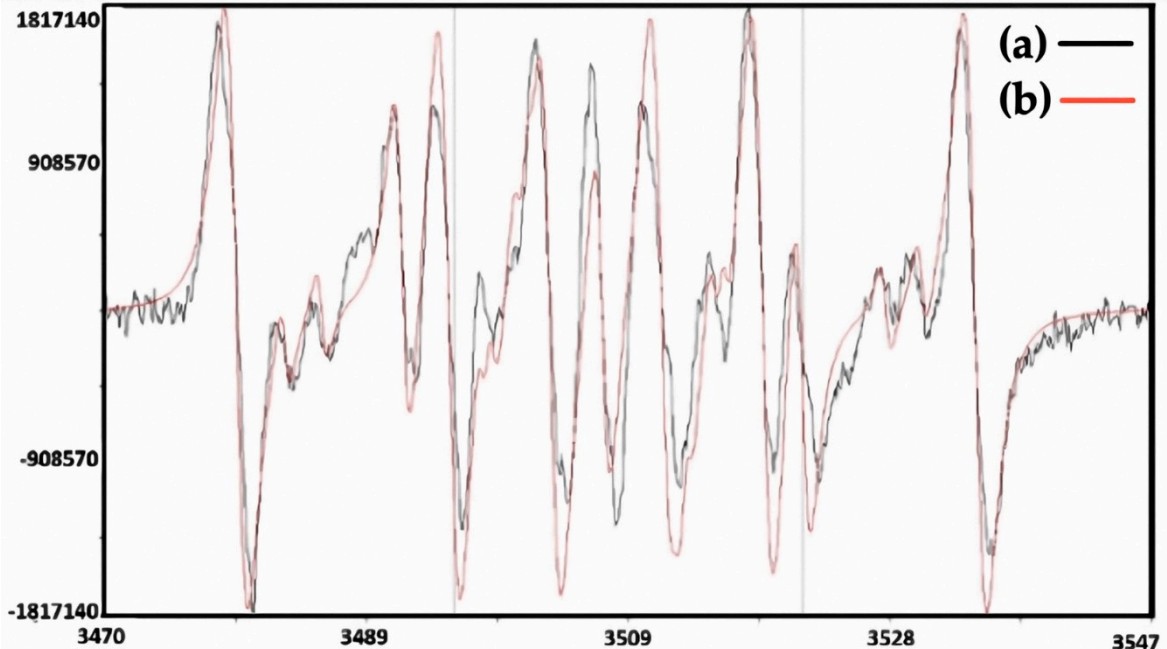

**Figure 10.** (**a**) Experimental and (**b**) simulated EPR spectra of DMPO-OOH adducts observed in the presence of the 800 °C nanocatalyst in phosphate buffer 7.4.

**Table 5.** EPR parameters in phosphate buffer for simulated spectrum by WinSim.

| Adduct | Hyperfine Splitting Constant (G) | | | Percentage |
|---|---|---|---|---|
| | $a_N$ | $a_{H\beta}$ | $a_{H\gamma}$ | |
| **500 °C** | | | | |
| DMPO-OOH | 15.668 | 23.372 | 0.384 | 83.190 |
| DMPOX | 14.729 | - | - | 11.322 |
| DMPO-OH | 14.208 | 15.903 | 2.780 | 5.487 |
| **600 °C** | | | | |
| DMPO-OOH | 15.478 | 23.580 | 0.950 | 68.626 |
| DMPOX | 14.700 | - | - | 25.754 |
| DMPO-OH | 14.589 | 16.539 | 2.828 | 5.620 |
| **700 °C** | | | | |
| DMPO-OOH | 15.646 | 22.866 | 0.501 | 79.773 |
| DMPOX | 14.846 | - | - | 14.184 |
| DMPO-OH | 14.382 | 16.434 | 2.636 | 6.042 |
| **800 °C** | | | | |
| DMPO-OOH | 15.622 | 23.037 | 0.491 | 80.189 |
| DMPOX | 14.706 | - | - | 14.013 |
| DMPO-OH | 14.406 | 15.302 | 2.685 | 5.797 |
| **900 °C** | | | | |
| DMPO-OOH | 15.631 | 23.324 | 0.466 | 82.237 |
| DMPOX | 14.753 | - | - | 12.835 |
| DMPO-OH | 13.984 | 15.762 | 2.717 | 4.927 |

ESI–MS of methylene blue decomposition was performed to check the possible structures of decomposition products before and after the leaching test as shown in Scheme 2. Before the leaching test, it was observed at m/z of 284 which was attributed to original MB. The new results appeared at m/z of 301, 221, 161, and 149 after leaching (5–125 min) indicate that MB was decomposed due to the breaking of the MB molecule [73,74]. The detected low molecular weight species confirmed the decomposition of methylene blue in the presence of iron oxide/carbon nanocatalysts. The mechanism of methylene blue decomposition in the presence of iron-oxide/carbon nanocomposites may be due to the attack of hydroxyl radicals as proposed in the previous report [18,68,75]. This nanocatalyst was attractive because it provided reusability for decomposition of methylene blue up to 12 times without loss of catalytic activity as shown in Figure 11. In addition the catalytic activity in the second cycle became more highly active than the first cycle due to high crystallinity of nanocatalysts. The nanocomposites showed the leaching of iron into solution at 21.6 mg/dm$^3$ when checked by atomic absorption spectroscopy. This system showed the comparable leaching of iron species to the amorphous catalyst in other reports [76,77].

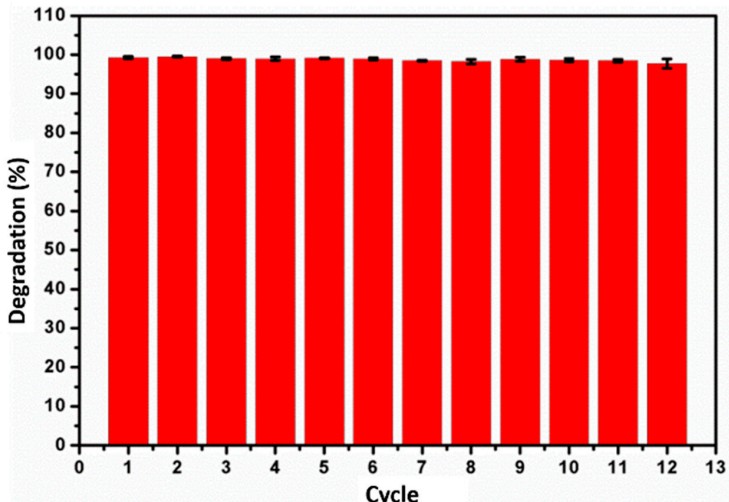

**Scheme 2.** Electrospray Ionization–Mass Spectrometry (ESI–MS) of methylene blue after decomposition by the 800 °C catalyst at reaction time (**a**) stock methylene blue solution $1 \times 10^{-5}$ mol dm$^{-3}$, (**b**) 5 min, (**c**) 15 min, (**d**) 25 min, (**e**) 125 min.

**Figure 11.** The recycling of the 800 °C nanocatalyst in the decomposition of methylene blue.

## 3. Materials and Methods

### 3.1. Chemicals

Ferrocene was procured from Acros organic, sulfuric acid was used by BDH chemical Ltd., and methylene blue (MB) was from Merck.

### 3.2. Preparation of Nanocomposites

Ferrocene 3.0 g (0.016 mol) was mixed with 2.5 cm$^3$ of concentrated sulfuric acid and the mixture was stirred until the color changed to dark blue. The mixtures were first heated at 100 °C for 1 h

and followed by at a specific temperature (400, 500, 600, 700, 800 and 900 °C) for 5 h in a Coor$^{TM}$ high-alumina crucible, capacity 100 cm$^3$.

### 3.3. Decomposition of Methylene blue

The catalyst at 25, 50, 100, 150, and 200 mg of 500–900 °C was added into the methylene blue solution $1 \times 10^{-5}$ mol/dm$^3$ (methylene blue dissolved in DI-water) in a 100 cm$^3$ beaker made from Pyrex glass. $H_2O_2$ is not necessary. The methylene blue was continuously stirred for 5 min in the dark for adsorption-desorption equilibrium. The methylene blue was sampled at the first 5 min for 3 cm$^3$ and poured into a centrifuge tube, and then centrifuged at 3300 rpm for 7 min. After sampling the MB at the first 5 min it was continually stirred. The MB was sampled every 10 min until 120 min and then centrifuged. After that, the MB solution was measured by UV-VIS spectroscopy from 400–700 nm (run DI-water as blank). Finally the MB solution which was sampled at the times of 5, 15, 25 and 125 min was characterized by ESI–MS. Moreover, the reusability of the catalyst was investigated. The percent of decomposition was defined as the Equation (1) where $A_0$ and $A_t$ were absorbances at the starting point and a specific time:

$$\text{Percent decomposition} = [(A_0 - A_t) \times 100]/A_0 \tag{1}$$

### 3.4. Instruments

Transmission electron microscopy (TEM) was performing using FEI, TECNAI T20 G2 Acc. Voltage 160 kV. The chemical composition of the sample surface was investigated by X-ray photoelectron spectrometer (XPS; AXIS ULTRADLD, Kratos analytical, Manchester UK.) The base pressure in the XPS analysis chamber was about $5 \times 10^{-9}$ torr. The samples were excited with X-ray hybrid mode $700 \times 300$ μm spot area with a monochromatic Al $K\alpha$ 1,2 radiation at 1.4 keV. X-ray anode was run at 15kV 10 mA 150 W. The photoelectrons were detected with a hemispherical analyzer positioned at an angle of 45° with respect to the normal to the sample surface. X-ray diffraction spectroscopy (XRD) was performed on Rigaku, Miniflex II, Japan. Thermal gravimetric analysis (TGA) was performed on METTLER-TOLEDO, model SDTA 851. Vibrating sample magnetometry (VSM) was performed on Lakeshore, 7404. Electron paramagnetic resonance spectroscopy (EPR) was performed on JEOL, JES-RE2X. A catalyst (3.04 mg) was dissolved in phosphate buffer solution (prepared by mixing $Na_2HPO_4$ 120.6 mg and $NaH_2PO_4$ 51 mg in deionized water 25 cm$^3$) 2 cm$^3$ and DMPO 5,5-Dimethyl-1-pyrroline $N$-oxide (7.637 μL), sonicated for 10 min and then pipetted into an EPR tube. EPR signals were simulated by Winsim program. UV-visible spectroscopy was performed on Jasco Model V-530. BET was performed by Autosorb-1, Quantachrome. CHN elemental analysis was performed by Perkin Elmer 2400 Series II CHNS/O Analyzer. ESI–MS was performed by Bruker MicroTOF. FAAS was perform on Perkin Elmer AA 3310, using Fe lamp (wavelength = 248.3 nm), flow rate 4 ml/min, and prepared standard Fe at 1, 2.5, 5, and 9 ppm (standard Fe 1000 ppm prepared from Fe (Cl$_2$), $6H_2O$ amount 4.9398 g in 20 cm$^3$ in concentrated $HNO_3$, diluted with $H_2O$).

## 4. Conclusions

Iron oxide/carbon nanocomposites were successfully prepared from the calcination of ferrocenium. This method was simple without sophisticated instruments such as a closed reactor and an inert atmosphere. Then the decomposition of methylene blue was tested for the catalytic activity of iron oxide/carbon nanocomposites. The decomposition of methylene blue was achieved by trapping molecular oxygen as hydroxyl radicals surrogate without UV irradiation. The active species were superoxide radicals derived from iron-oxide/carbon nanocomposites. This method can be applied in the water treatment without the requirement of added hydrogen peroxide.

**Author Contributions:** Investigation, formal analysis T.P. and T.T.; validation, P.C.; verification of research output, I.Y., T.M., and M.N.; supervising responsibility for valuable ideas, S.P.; research administrator, E.S.

**Funding:** This research received no external funding.

**Acknowledgments:** The financial supports by the Biofuel development for Thailand fund through Center of Excellence for Innovation in Chemistry (PERCH-CIC), Royal Golden Jubilee Ph.D. Program (Grant No. PHD/01242556 to PC, the Thailand Research Fund (RSA6080010), and the Office of the Higher Education Commission-Mahidol University under the National Research University Initiative are acknowledged.

**Conflicts of Interest:** The authors declare no conflict of interest.

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
