# Peer review of "Facile Synthesis of High Performance Iron Oxide/Carbon Nanocatalysts Derived from the Calcination of Ferrocenium for the Decomposition of Methylene Blue"

_catalysts, doi:10.3390/catal9110948_

Round 1
Reviewer 1 Report
Attached below.

Author Response
First of all, Tables 1-5 are not included in the manuscript. It is hard to discuss the results.
Tables 1-5 have been added.
In XRD patterns (Fig. 1), the clear peak is observed at around 18° for the samples calcined at 500, 600, and 700 ° This peak should be assigned.
The XRD peak at around 18° is probably an artefact from the measurements.
I think it is better to show XRD of the sample calcined at 400 °
The XRD of the sample calcined at 400 °C has been added.
The authors stated that the increase in calcination temperature (400 to 700 °C) results in the decrease of the particle size (Fig. 2). I cannot understand why the particle size is reduced by calcination at higher temperature.
It is probably due to the lower pencentage of carbon in the sample calcined at higher temperatures. The less number of aggregated particles was observed for the sample calcined at high temperature.
Is it possible to discuss the morphology (e.g. diameter) of each component of nanocatalysts (iron oxide and carbon)?
It is difficult to differentiate each component at this stage.
Reviewer 2 Report
Research is so good but I worry about some results.
Is it possible for 40 nm-sized hematite nanoparticles to have such a large magnetism? Please cite previrous report. hat does XRD peak of sample prepared at 400 ~ 800 oC mean at 18 degrees? In Figure 2b~d, I can not know what is major product. In addition, please inser HRTEM images of sample to show they have crystalline structure Can author provide size calculated by debye equation using XRD data? It would be helpful to confirm that the nanoparticles are single crystal. In the catalytic study, of course, using the same amount of sample yields the best results for organic-free sample at 900 oC. Author should experiment with the same amount of Fe.
Author Response
Is it possible for 40 nm-sized hematite nanoparticles to have such a large magnetism? Please cite previrous report.
In our report, it may be the mixture between carbon and iron oxides that make the anomalous magnetic property. In our previous report, radicals were detected in the ferrocene-derived nanocatalysts. Those radicals with iron oxides may exhibit this magnetic property.
hat does XRD peak of sample prepared at 400 ~ 800 oC mean at 18 degrees?
It is probably an artefact.
In Figure 2b~d, I can not know what is major product.
It is difficult to differentiate the products in the picture. From other experiments, the lower percentage of carbon was detected at higher calcination temperatures.
In addition, please insert HRTEM images of sample to show they have crystalline structure
Most of our samples are amorphous. It is hard to have the HRTEM images by this time.
Can author provide size calculated by debye equation using XRD data? It would be helpful to confirm that the nanoparticles are single crystal.
This part has been added.
In the catalytic study, of course, using the same amount of sample yields the best results for organic-free sample at 900 oC. Author should experiment with the same amount of Fe.
This experiment has been done. The lower activity was observed.
Round 2
Reviewer 2 Report
Response from author is reasonable. Therefore I think it can be published.
Thank you very much.